# Appealing Renewable Materials in Green Chemistry

**DOI:** 10.3390/molecules27061988

**Published:** 2022-03-19

**Authors:** Federico Casti, Francesco Basoccu, Rita Mocci, Lidia De Luca, Andrea Porcheddu, Federico Cuccu

**Affiliations:** 1Dipartimento di Scienze Chimiche e Geologiche, Università degli Studi di Cagliari, Cittadella Universitaria, Monserrato, 09042 Cagliari, Italy; federico.casti@unica.it (F.C.); basoccufrancesco@gmail.com (F.B.); rita.mocci@unica.it (R.M.); 2Dipartimento di Chimica e Farmacia, Università degli Studi di Sassari, Via Vienna 2, 07100 Sassari, Italy; ldeluca@uniss.it

**Keywords:** green chemistry, organic synthesis, wool, silk, feathers, plants derivatives

## Abstract

In just a few years, chemists have significantly changed their approach to the synthesis of organic molecules in the laboratory and industry. Researchers are encouraged to approach “greener” reagents, solvents, and methodologies, to go hand in hand with the world’s environmental matter, such as water, soil, and air pollution. The employment of plant and animal derivates that are commonly regarded as “waste material” has paved the way for the development of new green strategies. In this review, the most important innovations in this field have been highlighted, paying due attention to those materials that have played a crucial role in organic reactions: wool, silk, and feather. Moreover, we decided to focus on the other most important supports and catalysts in green syntheses, such as proteins and their derivates. Different materials have shown prominent activity in the adsorption of metals and organic dyes, which has constituted a relevant scope in the last two decades. We intend to furnish a complete screening of the application given to these materials and contribute to their potential future utilization.

## 1. Introduction

Nowadays, it is only natural to think in perspective to our planet to respect and safeguard it in each of its components. Global warming, melting glaciers, air, water, and soil pollution are just some of the problems we have been subjected to for several years; each issue has its roots in various causes, sometimes closely related one to the other. Providing for a common cause, it is correct to assert that neglect or underestimating the environmental issue has inevitably led to the current situation. Not only has this fact increased ecological risk, but it has also dramatically raised the risk of acute and chronic toxicity for living beings. The scientific community has recently decided to land in safer harbors; those of eco-sustainability and chemistry are obviously not excluded from this context. On the contrary, it has become its spokesperson, emerging with the branch of green chemistry. The first conceptualization is traced back to the early 1990s and became a widespread field in 1998 with the establishment of its basilar 12 principles [1]. A green approach means thinking outside of the box, leaving behind a whole series of certainties, such as using toxic solvents that guarantee the success of the reactions and finding, however, an equally effective way to replace them. Moreover, when the risks of a chemical reaction are assessed, the operator must take into consideration that each substance has a degree of intrinsic danger. Unfortunately, most of the reactions require the use of catalysts, which can be of various kinds: acidic, basic, metal-based, etc., and each of these has its own inherent risk, which could be addressed to the human body and even the ecosystem as a whole. Metal-based catalysts are hazardous compounds. the toxicity of which strictly depends on the metal they are composed of (Figure 1) [2].

For this purpose, one of the most important and studied fields in chemistry has always been the removal and the adsorption of pollutants present in the environment as by-products of many industrial and non-industrial processes. Metals and inorganic/organic compounds can cause a wide range of pathologies, such as neurologic, oncologic, and pulmonary diseases [3,4,5]. Over the years, various methodologies and materials have been developed and used in this field to lessen the environmental impact. Zeolites, carbon, peat, and many other materials have proved to be particularly effective in eliminating pollutants from soil, air, and waters [6,7]. Furthermore, many electrochemical processes could be used as well [8,9].

At the same time, an emergent field in chemistry has taken hold, enhancing natural raw materials, such as wool and feathers. These materials, derived from industrial and livestock processes, have few applications and are often difficult to dispose of, rendering the recyclability and recovery of these materials appealing for industries and producers. In the last two decades, natural products have been thoroughly examined and tested in the context of green chemistry. They gave very interesting feedback; wool and feathers have been intensely studied in the field of metal and dyes adsorption. Intriguingly, a similar response was given by fruit peel and some roots. 

Generally, supports for catalysis are made of metal oxides and silicates, which, besides being in many cases very toxic compounds, are largely sensitive to temperature, moisture, air, and some particularly severe reaction conditions. Catalyst poisoning is a dramatic condition because it does not allow the reuse of the catalyst, thus involving a significant economic outlay and increased environmental risk. For instance, zeolites used as support for organic reactions suffer the irreversible chemical adsorption of organic products and by-products, becoming less efficient. At the same time, carbon is too expensive and requires very high temperatures to be activated. 

Wool, silk, and the other natural derivatives gave excellent responses, even as catalysts for organic reactions, giving alternative and greener synthetic pathways.

A further advantage is the promotion of green solvents because they do not harm the chemical nature of these natural fibres, which are sensitive to traditional reaction conditions. Although performing well, they often require tedious pre-treatment to become competitive with current methods.

This review aims to underline the main benefits of green materials, focusing on their ability to adsorb dyes and pollutants and promote synthetic chemical processes, highlighting the specific properties of each of these elements.

## 2. Wool

As a result of industrial growth and the exploitation of chemical research worldwide development exploitation, the environment has been exposed to continuous risks and pollution: water, soil and air suffer the discharging of by-products such as metals, dyes, pigments, and toxic organic compounds. As previously reported, many materials and techniques have been applied to this field. Because of the importance of this subject, any kind of material has to comply with specific characteristics that match the canons of green chemistry: efficacy, low price, recyclability, very low/absence of toxicity, eco-compatibility. Besides these features, an adsorption material should show some peculiarities. Since wool, feathers, and the other natural products under consideration are mainly composed of proteins, their amino acid residue offers a good condition for building complexes with the positive charge that generally accompanies metals. Moreover, this amino acidic residue provides suitable shape complexes with the positive charge that accompanies metals typically.

Wool is a natural fiber obtained by the shearing of sheep. It is exceptionally highly versatile, mainly used by the construction, textile, and biomedical industry [10,11]. The main constituents of wool are keratinized proteins and amino acids, as well as, in smaller quantities, other organic compounds; fibers can vary a lot in length (20–300 mm), diameter (8–70 µm) and width (for Merinos is around 15–17 µm), in close dependence of the breed. Due to its peptidic composition, wool has an isoelectric region calculated in the range of 4.5–5 pH values [12]. This means that almost every carboxylic and the aminic group is present in its ionized form and even the contribution derived from the aminoacidic ionizable side chain is balanced. It follows that, at higher pH values, we have a predominance of negative charge contribution, while at lower ones, positive charges are prevalent; in reason of this, in terms of cationic bindings, the best-operating conditions are above the isoelectric point. It is self-sufficient that negatively charged wool works better on metal binding. Next to these optimal parameters, the best results for ion binding were given by a sharp increase in the surface area of the material. In 2014, El-Sayed et al. published an article about removing copper and zinc performed with wool micro powder [13]. In their work, they describe the treatments on the fibers that are required for their scope. After a fast clean-up of threads with water, sodium carbonate and non-ionic detergents, wool was treated with two different procedures: at first, wool fibers were oxidized with a mixture of H_2_O_2_ and TAED at 80 °C for 30 min and then washed with water and dried overnight (OW). Then, a portion of the OW and the not-oxidized wool were milled separately with a ball mill to obtain, respectively, oxidized wool powder (OWP) and wool powder (WP). These fibers were then tested to understand their potentiality in the uptake and release of some of the most common ions present in wastewaters. Substantially, the two plots below (Figure 2) are reported to show the affinity for two ions: Cu^2+^ and Zn^2+^.

These results give important insight into the experiments. With a closer look at the data, it is revealed that OWP has the best performance in the adsorption/desorption of the two ions, probably because of the high presence of sulphonic and carboxylic acids in the structure. Furthermore, the fact that the “powder form” of wool can better complex ions is considerable and most likely is closely related to the increase in the fibers’ surface area, which now exposes their functional groups in a fiber’s more significant amount. To partially validate this point, a 2012 study showed that through the air jet milling technique, the surface area could increase over 700 times compared with the related fiber [14]. So, all is governed by the charge; the pH value is a mandatory condition that must be respected for ion exchange, to operate in the optimal conditions for the capture and the release. Adding or removing protons can affect the formation of a complex.

Nonetheless, we could ask ourselves: in the diametrically opposite situation, could wool be used in the adsorption of anionic species? The answer lies in Wen’s work [15]. To obtain a general picture of the context, it is enough to think about what happens in the textile industry: the great amount of wool-containing textiles and everyday products are not used as they are, but very often undergoes a series of processes, one of which is the addition of dyes (coloring process). Dyes, very often, are organic molecules, such as diazo compounds, which are sufficiently conjugated to absorb light and reflect a color, but precisely because of their structure, they are also hazardous chemicals [16,17,18]. To remove this pollutant from water, Wen et al. used wool as an adsorbent in their work. C.I. Acid Red 88, C.I. Acid Red 18 (Figure 3) and Lanasol Blue CE were used as dye agents in a pH 4.5 solution (citric acid/disodium hydrogen phosphate).

The study compared the activity between wool and activated charcoal, mostly used for adsorption. Four types of wool were used: a fibre sized form, a fine cut one, and two different types of milled form. It resulted that the two not-milled forms could only adsorb low concentrations of dyes due to the natural cuticle which covers the fibre. Instead, when milled at a different size, absorption increases inversely proportional to the size of the powders. Indeed, suppose one thinks about the above-mentioned coloring process carried out on wool in the textile industry. In that case, this occurs at boiling temperatures, precisely to ensure the penetration of the dye through the hydrophobic and almost waterproof cuticle. The other side of the coin concerns the application this material finds in different branches, such as heterogeneous catalysis in organic chemistry. Such a type of catalysis is strongly affected by the toxic nature of the mostly known catalyst, which often suffers a rapid decay process, an issue that strongly limits its use in subsequent reactions [19]. Instead, by giving a quick look at the relevant inherent literature, we find out that the greener approach is a rising interest topic, as evidenced by the number of publications on the subject [20]. In a very detailed review [21], the first thing that catches the eye is that wool powder proved not only to be a valuable tool for catalysis itself, as a specific source of keratin, but also a complexing agent for metal compounds of relevance in catalysis. Offering some examples, we introduce an innovative work released in 2016 [22], which confers to wool powder the role of support for nitroaldol reactions. The idea is quite simple: keratin deriving from wool fibers (unfortunately no detailed treatments are reported), in the presence of a mixture of H_2_O/TBAB, can afford nitroaldol products in high yields (Figure 1), perfectly stackable with results obtained with other polymers and comparable with other previous experiments, which involve the use of DMSO as a solvent. Water remains the best choice for environmental issues related to DMSO’s toxicity. Still, it lacks in terms of reaction times (48 h instead of 24 h), probably due to the less solubility of the reagents.

In addition, another parameter should be considered: the recyclability of the material. Despite giving good results at its first use, the employment of DMSO drastically reduces the reusability of keratin. Instead, when the reaction is carried out in the water in the presence of TBAB, the reuse of the catalyst for at least three cycles is guaranteed with totally comparable results in terms of yields. This suggests that the choice of the solvent plays a crucial role in the life span of the material. 

Moreover, tremendous advances have been made in catalysis. Metals have contributed predominantly in the context of C-C bond formation [23,24,25] Hydrogenations and hydrations have also been thoroughly studied, [26,27] so intensively that it would be impossible to report them all in this review. Still, on this topic, the review mentioned above [22] comes to our rescue, proposing us a series of examples reported in the literature about new wool-mediated approaches. Due to the high content of cysteine, these fibres can efficiently complex transition metals, especially palladium. The shared interest in such an element is due to its wide use in organic reactions: are worth mentioning hydrogenations [28], several hydrations [29,30,31] and Suzuki coupling [32].

Wool property to adsorb metals is relevant, but even more so is gas retention, as in the case of CO_2_ and the ability to make it reactive. While the capability of complexing metals is attributed to the thiol groups, aminic residues are able to efficiently trap CO_2_. Additionally, this precise characteristic aroused the interest of Chang et al. in their work [33]. Carbon dioxide can be an extremely versatile tool if correctly activated; the only problem is that we are dealing with a basically stable and poorly reactive compound. However, when the energy gap is bridged, a series of valuable reactions can be accessed. In the specific case of this work, the authors focused their attention on epoxide cycloadditions with CO_2_ (Figure 2), carrying out a well-defined solvent-free green procedure applicable to a wide range of epoxydic substrates. The cornerstone of this technique is defined by the combination of keratin and a co-catalyst (Table 1) (more specifics of the reaction can be found in the work corresponding to Reference [33]).

We also underline the highly effective C-C bond coupling procedure, thoroughly described in 2012 by Lei and co-workers [34]. In their paper, the authors demonstrate the feasibility of conducting a Suzuki Pd-catalysed reaction under mild and green conditions, also guaranteeing recyclability of the biopolymer complex (PdCl_2_-Wool) at least three times, almost without any metal leaching or other activation steps. To perform standard Suzuki coupling, harsh conditions are usually required, such as toxic solvents (toluene, dioxane, DMF, THF) [35,36] and diverse types of ligands for Pd complexation (phosphine and heterocyclic compounds). In this case, wool acts as a complexing agent for palladium thanks to the thiol residues, thus allowing the use of the least impacting solvent of all, i.e., water, and a catalytic amount of K_2_CO_3_ as a base. This technique proved to be applicable to a broad range of substrates, but other critical parameters must also be considered. One of the most essential is the recyclability of the complex, which is always sought in heterogeneous catalysis. Therefore, two reference compounds from the scope were chosen (iodobenzene and phenylboronic acid), as is shown in Figure 4. Moreover, critical data was derived from the study of the solvent mixtures for the reaction. It is quite common for a Suzuki coupling to be performed in a binary solvent mixture, often composed of an organic solvent (THF or dioxane) and water, in different portions (9.5:0.5). What can be deduced by Lei’s study is that in water alone, reactions take place, giving very high yields. However, as THF is gradually added in higher portions, yields drastically decrease (Figure 5). The authors explained this phenomenon with the natural hydrophobic tendency of wool, which prefers hydrophobic interactions when exposed to water, in order to reduce the contact surface with the polar environment, which perhaps ensures the promotion of the transition state.

## 3. Feathers

Considering what was mentioned above regarding wool fibers, we can quickly move to another widespread and studied material: hen/chicken feather. The main constituent of this peculiar fiber is, once again, keratin. Biochemistry and biology tell us that proteins are one of the essential macromolecules which define life [37]. Among all the biochemical and physiological processes, these macromolecules create a thick protective layer against animals’ external and potentially dangerous stimuli. Valuable fibers, such as wool (described in the paragraph above) and feathers (current section), derive from this biological response. It is intuitive to think that the general composition is more than similar to wool: feathers, beyond sharing an animal origin and a keratin structure, present a very similar amino acid composition, only enriched in sulfur-containing side-chain amino acids [38]. This peculiarity, in addition to giving fibers greater strength, provides a good chelating property. Numerous studies have focused on pollutant removal, exactly as we previously saw for wool. Considering the previous studies, some articles captured our interest because they show similar approaches in evaluating material adsorption/desorption properties. For example, considering the paper by Al-Rousan published in 2002 [39], authors reported an important screening on adsorption competition between heavy metals, comparing the results obtained in a solution of a single ionic species with those of a binary blend. As we previously said, many industrial processes produce a discrete number of pollutants, most of which are metals. 

To perform the adsorption experiments, the authors used clean-cut chicken feathers, washed with water and detergent and dried for 2 days in an oven at 70 °C. The feathers were tested on metal salts mixtures composed of (CuSO_4_·5H_2_O)–(ZnSO_4_·7H_2_O), (CuSO_4_·5H_2_O)–(NiSO_4_·5H_2_O) and, at least, (ZnSO_4_·7H_2_O)–(NiSO_4_·5H_2_O). Solutions of these salts were prepared at different concentrations, and feathers were used in a known amount such that the resultant ratio was 5 mg/mL. The adsorption study lasted for 24 h at 25 °C in aqueous media. At the end of the experiments, feathers’ fibers were filtrated, and the solution obtained was analyzed by atomic absorption spectrophotometry. To evaluate adsorption features, the authors referenced three different isotherm models: Freundlich (Equation (1)), Langmuir (Equation (2)) and Sips’ ones (Equation (3)).
(1)$$q_{e}=k_{F}C_{e}^{\frac{1}{n}}$$
(2)$$q_{e}=\frac{q_{m}{\mathrm{bC}}_{e}}{1+{\mathrm{bC}}_{e}}$$
(3)$$q=\frac{K_{s}{\left(\mathrm{bC}\right)}^{\frac{1}{n}}}{1+{\left(\mathrm{bC}\right)}^{\frac{1}{n}}}$$

Differences between the three models have their origin from the fact that they take different specific parameters into account. The Freundlich method considers the equilibrium condition, which is a relationship between the not-adsorbed species and the adsorbed one through Freundlich constants (on account of sorption capacity and intensity). On the contrary, Langmuir’s model considers both the power and energy of adsorption by using Langmuir’s constants. Eventually, Sips’ isotherm model represents and describes adsorption with parameters derived from combining the two previously mentioned approaches. Fitting both predicted and empirical data, results can be summarized as follows: single ion adsorption experiments demonstrated that chicken feathers have a greater affinity with copper than the other two bivalent ions. At the same time, zinc is preferred to nickel, which has a worse interaction with feathers. Concerning the binary mixture experiments, it was found that ions were adsorbed less extensively than a single ion in solution, and this feature is even more noticeable when the concentration of one ion far exceeds the other. The last consideration that could be displayed from this paper is that the Freundlich model was found to better fit the empirical data than the other two methods (Table 2). 

As proof of the importance of these studies, many papers about metal sorption were published further in [40,41,42] about potentially toxic cations removal. Once again, the application of these materials is not limited to metal ions, but it is also efficient with organic compounds. We have already discussed the importance of this topic in the context of “greener” applications related to wool. Interestingly, the same considerations can be made on feathers, namely the problem of removing organic pollutants from wastewater [43,44,45]. A particular issue is the presence of tartrazine (Figure 6), an extreme dangerous dye used for various purposes in the food industry and cosmetics. This compound is reported to cause a wide range of severe diseases and pathologies, among which we find asthma, migraines, cancer, etc. [46,47]. In 2007, Mittal’s group published an article about removing this dye from wastewater with the employment of hen feathers [48], whose fibres were washed with distilled water, dried, cut with scissors and used after 24 h of pre-treatment under oxidizing conditions (H_2_O_2_).

As we previously reported for metal adsorption, Langmuir and Freundlich’s models were used to obtain the adsorption profile compared with the experimental data. Various parameters were tested, among which we found interesting the trend between the amount of absorbed dye and different amounts of absorbent at three different temperatures (30, 40 and 50 °C) and pH/removal plot at pH range from 2 to 6. (Figure 7 and Figure 8).

Considerations on the plots are mandatory. First, data derived from different temperatures shows how the adsorption increases proportionally with rising temperatures. As the authors also stated, this suggests that we are facing an endothermic system. Another consistent fact is that by increasing the number of feathers, the adsorption process is improved too: this means, in terms of kinetic energy, the rate is dosage-dependent on the material used. It follows that the rate of adsorption increases. Finally, pH plays a crucial role in the adsorption process, as the best condition to operate was found to be a pH value of 2. Above all, feathers proved to be helpful in dyestuff removal, pointing out the potential application for their green use. Although the easiness and simpleness of the experiments, the authors demonstrated the strength of this methodology, which fits very well with both Langmuir and Freundlich models and several kinetic and thermodynamic parameters. 

Due to its keratin structure, the feather is a suitable substrate for heterogeneous catalysis and works very well in hydrogenation reactions [49], heterocycle click synthesis [50] and organics oxidations [51]. Nevertheless, little is known about reactions involving feathers as a catalyst. C-C bond formation remains a crucial issue, mainly due to its application in various synthetic protocols, and Suzuki coupling represents the leading procedure above all the others. For this reason, and thanks to fibre’s ability to chelate transition metals, Jain et al. developed an innovative synthetic strategy by employing Fe_3_O_4_@/CF-Pd(0) complex in Suzuki coupling reactions [52]. They report the catalyst’s synthesis, which contemplates an easy and rapid treatment of magnetic nano-ferrite (Fe_3_O_4_) with chicken feathers in powder and a palladium salt, subsequently reduced to obtain the catalyst itself. The reaction was tested on differently substituted aryl bromide (2 mmol) in the presence of aryl boronic acid (3 mmol), a base (K_2_CO_3_, 2 eq.) and the magnetic catalyst (0.5% mol of Pd) in water (Figure 3).

What immediately catches the eye is the efficiency of the reaction. In fact, products were provided in very high yields for both electron-withdrawing and -donating groups. Unfortunately, the same reaction with 4-chlorophenyl bromide provided a 34% yield despite lengthening reaction times (3 days). In addition to a series of optimization tests involving the catalyst’s concentration, amount and type of the base, reaction time, and temperatures, one of the most critical results derived from recycling tests: this type of catalyst showed to be sufficiently stable to allow six runs of the reaction without affecting any yield, confirming its value as a heterogeneous green catalyst.

## 4. Silk

The second-last topic of this review focuses on another animal derivative: silk. [53] In nature, several insects produce this material, but the most commonly known is *Bombyx mori*, from which cocoon silk is extracted. It is mainly composed of proteins, particularly fibroin [54], a β-keratin similar to those present in the human body. Fibroin is composed of β-sheets in a (Gly-Ser-Gly-Ala-Gly-Ala)*_n_* sequence; amino acids confer a general isoelectric point around 1.4–2.8 pH values [55,56,57].

These properties ensure the optimal theoretical parameters for pollutant removal, especially metals [58,59]. Nevertheless, silk has been little researched and used as an adsorbent material. In 2021, Mia et al. contributed to this field with a very interesting paper about removing nickel, cadmium and copper from water [60]. Little modifications to silk fibers (SF) were implemented, such as the doping with dopamine, because the neurotransmitter, due to its catechol group, can complex metals as well. To perform this pre-treatment, silk was first immersed into a solution of tris(hydroxymethyl) aminomethane (50 mmol/L) and dopamine (2 g/L) for 24 h at 30 °C and constant pH of 8.5. Eventually, it followed some washing and a drying step.

Data were collected by considering three different parameters: the initial quantity of metals, the amount of silk-adsorbent, and the experiments’ lasting, all combined with prediction models based on Freundlich and Langmuir isotherms. 

Regarding the first study, the authors decided to consider different initial concentrations of ions; five solutions (20/40/60/80/100 mg/L) of the various other metals were prepared, and 400 mg of adsorbents were designed and added inside for 60 min at 30 °C. The data established that the maximum adsorption is not above 60 mg/L (Figure 9).

By these assumptions, the following experiments were conducted with 60 mg/L concentration to keep constant the first verified maximum of removal. For this purpose, the adsorbent dosage was tested by slightly increasing its amount in the solution (0.2/0.4/0.6/0.8/1.0 g of SF). The graphics below show the trend for ions removal on account of SF mass (Figure 10).

As evidenced from Figure 10, a difference has been observed: while cadmium ions reached the maximum adsorption with a total of 0.4 g of adsorbent, copper and nickel required a 0.6 g dosage to be wholly removed from the solution. However, since even Cu^2+^ and Ni^2+^ were adsorbed in very high percentages (99% and 94%, respectively), balancing economic/efficiency parameters, the optimum was considered 0.4 g of SF.

With these two optimized parameters in hand, the maximum lasting of the experiments was evaluated. From the curves described in Figure 11, it can be easily displayed that just 10 min was sufficient to entirely remove Cd^2+^ and 15 min for the other two ions, at 30 °C. 

The following conclusions can therefore be drawn. Considering all the data, the authors proved the efficacy of SF in heavy metals adsorption, shedding light on which metals is preferred (Cd^2+^ > Cu^2+^ > Ni^2+^), and provided the efficiency parameters for their conditions. Moreover, it was found that Langmuir model fitted better the results than Freundlich’s, and the adsorption kinetic is better described by a sort of second-order trend. 

In terms of catalytic activity, many reactions have been reported until now. Noteworthy is a very recent Suzuki-Miyaura coupling by Farinola et al. [61], who proposed an innovative procedure mediated by a new Pd-silk fibroin catalyst that shows enhanced activity compared to other catalysts, and a long lifespan. Their last work [62] on Suzuki-Miyaura and Ullmann reactions, where they developed new synthetic procedures involving chlorobenzenes, whose reactivity is much lower than the iodine counterpart, supported a Pd-silk fibroin complex, which is reusable for 25 times without any drawbacks (Figure 12). 

Moreover, similar catalytic systems have been studied in other reactions. For instance, heavy metal adsorption on silk is reported to occur with different methodologies [63,64,65]. This feature allows the preparation of catalysts useful for selective hydrogenations since other catalysts lack selectivity. 

Hirota and co-authors worked in this direction, developing an efficient and newsworthy protocol for selective reduction of alkene compounds [66]. In their paper, catalyst synthesis involves the soaking of fibroin in a methanol-Pd(AcOEt)_2_ solution at room temperature, which provides the reduction of Pd (II) to Pd (0), probably favoured by the participation of methanol in the redox process. As a result, authors reported the reduction of olefine moiety without affecting the carbonyl group, whilst the classic Pd/C-mediated hydrogenations conducted on ketones and aldehydes yielded the related alcohol or alkane, depending on the H_2_ stoichiometry. Conversely, when a ketone was subjected to reductive silk-Pd-complex conditions, the reaction did not occur.

With this surprising data in hand, Hirota’s team explored the reaction scope for their catalyst (Figure 4).

Noteworthy is the fact that the reduction selectivity is also strongly influenced by the nature of the solvent used; indeed, the best results are provided by AcOEt, which does not affect the carbonyl group of aromatic aldehydes, while a variable amount (from 10 to 33%) of the corresponding alcohol was detected when THF, DCM and MeOH had been used. The authors ascribe this behaviour to the intramolecular coordination complex generated between Pd, olefin and aldehyde, which is disfavoured by the AcOEt fashion of being a better ligand than aldehyde.

Aside from smaller studies on carbonyl group and aryl halide reduction themselves, authors focused their attention on functional group competition, such as halides and olefins, as shown in Figure 5.

Other compatibilities were tested, for instance, olefins reductions and esters’ (benzyl) and amines’ (Cbz) deprotection, demonstrating that the methodology is selective towards olefins and does not affect the protecting groups.

## 5. Plant Derivatives

A pivotal role in such a scene is played by the industrial wastewaters, which are plenty of inorganic and organic pollutants. This is the reason why the industrial processes started remodelling and reconsidering their activities in a greener fashion.

Traditional methods used to remove pollutants from aqueous solutions include chemical precipitation, membrane separation processes, biological degradation, oxidation, solvent extraction, and eventually adsorption [67,68]. As we mentioned in the other paragraphs, researchers have only recently turned their attention towards new methods involving the use of new adsorbents to remove pollutants from wastewater [69]. These non-conventional adsorbents are classified into five categories: waste materials from agriculture and industry, fruit waste, plant waste, natural inorganic materials, and bio adsorbents (living and non-living biomass) [70].

Some of the waste materials have already been discussed, thus we turn to the last class under consideration: Fruit Peels Waste (FPW).

The large availability of fruit peel waste (Table 3), combined with its meagre cost, has prompted the scientific community to consider it a potentially exploitable material for the adsorption of dyes and metals [71].

Among all pollutants, the organic ones predominantly have good solubility in water but with poor biodegradation properties. This mainly involves acids, such as carboxylic and sulphonic acids, phenolic compounds, and dyes, visible even at very low concentration [72,73,74], pesticides, [75] oils, and others [76].

Above all the others, phenols are considered persistent pollutants on account of their accumulation properties in the environment and in the organism. For this reason, the WHO has recommended that the phenol content in water should not exceed 0.02 µg/L [77]. To limit their concentration, cheap adsorbents such as resins, both natural and synthetic, are commonly used [78]. 

In Table 4, a comparison between the most relevant application and experiments is provided [79,80]. 

Orange and banana peels show enhanced adsorption properties. OP exhibits a higher affinity for ions and dyes than the other citrus derivatives. At the same time, BP is the only versatile peel to remove pollutants of different kinds, including ionic and non-ionic organic substances. This limited selectivity could be attributed to some basic characters to which it owes its affinity with cations and anions [90] The OP surface can be modified by generating xanthates, forming complexes with metals [91]. Xanthates are formed by the reaction of carbon disulphide, under caustic conditions, with pre-existing hydroxyl substrates of the OP. Other studies proved that OP is adequately helpful in the removal of several organic pollutants, including 1-naphthol and naphthalene, even with pyrolysis treatment [70], or dyes such as Direct Blue-86 and Direct Navy Blue 106, again by pyrolysis of OP using 98% H_3_PO_4_ [92,93].

As mentioned before, the peels are composed of various types of fibres that can carry out the function of pollutants adsorption. Heavy metals are better adsorbed by fruits peels rich in pectin, which carboxylic groups are responsible for generating the complex [94,95].

The main parameters influencing adsorption capacity are the peels’ surface properties and the adsorbate’s chemical core. Agricultural wastes are mainly constituted by cellulose, hemicellulose, lignin, lipids, proteins, polysaccharides, and pigments containing different functional groups.

The surface area, the volume and the diameter of the pores are critical factors in such a process. The first one is determining the adsorbing capacity, but a lower surface area characterizes peels compared to carbon and siliceous materials. The volume and diameter of the pores determine the degree of diffusion of the polluting molecules through the FP. Other variable operating parameters could be temperature, the increment of which corresponds to greater mobility of the adsorbate, solution pH, and the contact time. Poor results could also be improved by varying the initial peel concentration, its particle size, or the initial adsorbate concentration.

To perform the maximum absorption, it is mandatory to select an appropriate activation process: this involves an initial cleaning and washing pre-treatment with solvents, an acid, or a base, drying and heating step of the FPWs. Then, the chemical treatment is used to improve absorption properties, such as the internalization of water, by increasing or decreasing hydrophilicity, the ion exchange capacity with it or the conductivity of the system: these methods typically include protonation, chemical xanthation and pyrolysis, saponification, or oxidation.

Other derivatives proved to be very effective in adsorption and removal of pollutants, particularly in soil, particularly plants, which exploit their roots for decontaminating it. *Rhizofiltration* is an intrinsic property of some plants, which use their roots to absorb water, nutrients, ions, and organic compounds from the soil. Chemists took their cue from this peculiar feature to purify soil and water by contaminants, especially metals and ions [96].

This is a versatile tool in catalysis, as recently reported Suzuki coupling roots-mediated demonstrated [97]. In this paper, the adsorptive properties of Eichhornia Crassipes, also known as water hyacinth, are meticulously underlined; its roots can complex metal species in swampy and stagnant sites, especially Pd, and, for this reason, it was used for Pd-Suzuki catalysed reactions. In addition to a very broad scope performed on heteroaryl and aryl compounds (Figure 6), a solvent screening and a catalyst loading study provided very interesting figures, as evidenced in Table 5 and Table 6.

In 2013, Grison et al. published a paper [98] about the synthetical application of two plants, *Psychotria douarrei* and *Geissois pruinose*, two metallophytes, a type of plant that grows in metal-rich soil. Due to this property, it was found that they were abundant in nickel, which is a versatile metal for catalysis, particularly useful in Lewis’s acid catalysis.

The authors analysed their plant’s samples appropriately, which showed a high mass percentage of nickel compared to other metals. They subsequently subjected their material to two different conditions: one material was not treated but directly used after dispersion in montmorillonite K-10. The other one was enriched in commercially NiCl_2_ and dispersed in montmorillonite too. When the two materials were used to perform Biginelli’s reaction, the results were somewhat discrepant, though natural and untreated plants gave the best results. The possible explanation could be the contribution given by the other metals naturally present in the material, which is probably inhibited by adding another source of nickel in the treated one. Grison’s team extended their reaction on different substrates, providing thirteen tetrahydro-pyrimidinic products (Figure 7).

With these results, the authors confirmed that their procedure can completely overlap with the literature data. Moreover, they obtained Biginelli’s products in higher yields and greener ways, compared with NiCl_2_ traditional catalysis. These are encouraging results for further investigations that could raise interest in phytocatalysis. 

## 6. Conclusions

In view of the above, with this collection of papers and pieces of knowledge on green methods and materials, we aimed to present a general overview of how green chemical processes can be terrifically performed. Great strides have been made in understanding the composition of these raw materials, in some cases considered a waste, now rethought as a medium for the extraction of pollutants the adsorption of metals and organic molecules for heterogeneous catalysis. With a deeper understanding of the mechanisms behind this adsorption process, the surface energies generated, and the chemical composition of these materials, predictions for organic reactions could be easily drawn. Even if such materials have already proved to be valuable choices for heterogeneous catalysis, a more intensive investigation of their properties could lead to further breakthroughs in metal catalysis. So, recyclability is without a shadow of a doubt the key element when a catalytic process is established, and the use of these natural products for such a purpose just skyrockets all the advantages brought by their use. 

Future outlooks involve discovering better conditions in which the eradication of toxic compounds is as fostered as possible. New catalytical processes can provide some well-known reactions and, perhaps, some new ones.

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
