# Peer review of "Appealing Renewable Materials in Green Chemistry"

_molecules, 2022, doi:10.3390/molecules27061988_

Round 1
Reviewer 1 Report
Herein, the authors review renewable materials in green chemistry: wool, silk, feathers, and fruit peel waste as alternatives to reaction media and environmental remediation. The review is well-written and the authors cover numerous references. A notable limitation of this review is that several different materials are covered which limits the ability of the authors to conduct a deep review of each material independently; however, it allows them to reach a broader audience.
Author Response
We thank the reviewer for recommending the publication of the manuscript. We appreciate his/her comments on the previous version, which assisted us in making corrections that improved the manuscript.
Reviewer 2 Report
This review deals with the appealing renewable materials in green chemistry emphasizing wool, feathers, silk, and the other natural derivatives as green materials in the applications of pollutants and dyes adsorption and alternative synthetic chemical processes.
In my point of view, this review discloses and compiles interesting applications of these green materials. However, some points have to be clarified before being considered for publication in Molecules. I would recommend this manuscript as a major revision.
The main comments are as follows:
Introduction
- The introduction part must be modified. Especially, on page 2, the authors could present, point out and compare clearly the advantage and disadvantages of the green materials presented in the manuscript to other conventional used materials. The consequence could certainly convince that the use of wool, feathers, silk and other natural derivatives is promising and interesting materials for adsorption and synthetic chemical processes.
- The authors present and discuss the sequence of each material ability following dyes and pollutants adsorption and synthetic chemical processes throughout the manuscript. After I have read repeatedly many times, in my point of view, this strategy is difficult to understand and follow with the details in the manuscript. I would recommend the authors rearrange by two main applications gathering data of dyes and pollutants adsorption of each material together. Also, in terms of synthetic chemical process applications, they should be modified in the same manner. This could allow other researchers to ease comparing and understanding the activity in catalytic reaction or adsorption capacity for those materials.
Author Response
We are grateful for the reviewer’s positive comments on the quality of the manuscript and on the viability of the current approach.
- As requested by the reviewer, on page 2 and in the full text, we have better reinforced the green features of these natural media compared to synthetic analogues.
“Generally, supports for catalysis are made of metal oxides and silicates, which, besides being in many cases very toxic compounds, are largely sensitive to temperature, moisture, air, and some particularly drastic reaction conditions. Catalyst poisoning is a dramatic condition because it does not allow the reuse of the catalyst, thus involving a significant economic outlay and an increased environmental risk. For instance, zeolites used as support for organic reactions suffer the irreversible chemical adsorption of organic products and by-products, becoming less efficient, while carbon is too expensive and requires very high temperatures to be activated.
Wool, silk, and the other natural derivatives gave excellent responses even as catalysts for organic reactions, providing alternative and greener synthetic pathways.
A further advantage is the promotion of green solvents because they do not harm the chemical nature of these natural fibres, which are sensitive to traditional reaction conditions. Although performing well, they often require tedious pre-treatment to become competitive with current methods.
- In drafting the manuscript, we considered a different rearrangement of the review, including the solution proposed by the reviewer, but this was less effective for the reader. Furthermore, a material-oriented organization helps the reader better understand the strengths of each of the different supports investigated. Although we understand reviewer 2's concerns well, since all other reviewers also agree with our organization, we have kept the organization of the original manuscript. We hope the reviewer understands; we certainly do not mean just to dismiss this thoughtful comment.
Reviewer 3 Report
The manuscript review entitled “Appealing Renewable Materials in Green Chemistry” from Casti et al involved the bibliographical analyses of several materials (wool, silk, feathers, plant derivatives) for novel green strategies regarding the utilization of them instead the use of toxic catalysts.
The introduction is according to the developed topic of the manuscript, and it has updated bibliographical references to support the research. Besides, it is important to consider several possible corrections/additions:
- It is necessary to explain the relationship between “waste material” phrase and the selection of t he materials the authors did for this manuscript (wool, silk, feather). Wool and silk are raw or final material from several industries.
- Line 35: Only recently phrase: in fact, the green chemistry approach is used from 1990. I suggest analyzing the origins of green chemistry to improve the introduction (https://greenchemistry.yale.edu/about/history-green-chemistry)
- Moreover, it is important to take into account different parameters/metrics to analyze the impact of metal-based catalysts and the materials of interest due to the potential use of them replacing the usual reagents/solvents/materials, throughout the manuscript. (Such as atom economy, reaction mass efficiency, E-factor, etc.)
Furthermore, I encourage the authors to check some mistakes (in yellow, pdf file attached) such as:
- Figure 1: involved instead invovled
- There are several blank spaces throughout the manuscript
- Please remember that the unit and the number (e.g. 20 nM – 50 °C – 10 min) should have a blank space between them
- Figure 3, Scheme 1 and table, Scheme 3, Figure 12, Scheme 4, Scheme 5, Scheme 6. Please check the correct size of these figures according to the guide authors (molecules and atoms look tinny at 100%-page size)
- Please homologate the size of the graphics throughout the manuscript
I would like to invite the authors to add the abbreviation list of words at the end of this manuscript.
Finally, I suggest improving the conclusions with the addition of a short sentence/paragraph suggesting/envisioning possible important applications that involve the use of these materials in the design of novel chemistry processes (replacing the use of catalysts).
I recommend the acceptance of this manuscript after the authors performed the suggested corrections/additions.

Author Response
We thank the reviewer for her/his constructive comments and valuable time for this work. Finally, we revised the paper on the review suggestion, going into more detail:
- The word “waste” has been replaced by “raw material”, a more appropriate expression.
- For point 2 raised by the reviewer, we have inserted reference 1 and the following statement: “The first conceptualization is traced back to the early 1990s and became a widespread field in 1998, with the establishment of its basilar 12 principles”.
- The authors, who are familiar with green chemistry metrics, have taken the excellent suggestion made by the reviewer into serious consideration. Unfortunately, many of the data needed to calculate the required parameters are missing from the original work, and a detailed comparison is not feasible. However, where possible, we have improved the green footprint of these natural materials; here is one of the sentences included in the text (Pg 8).
“We already discussed the importance of this topic in the context of “greener” applications related to wool. Interestingly, the same considerations can be made on feathers, namely the problem of removing organic pollutants from wastewater [ ]. A particular issue is the presence of tartrazine, an extreme dangerous dye used for various purposes in the food industry and cosmetics. This compound is reported to cause a wide range of serious diseases and pathologies, among which we find asthma, migraines, cancer, etc. [ , ]. In 2007, Mittal’s group published an article about removing this dye from wastewater with the employment of hen feathers [ ]….,
- We have fixed several mistakes in the full text, as suggested by the reviewer. We want to take this opportunity to thank her/him for his valuable work.
- As requested, a short sentence has been added to the conclusions to enhance them
Reviewer 4 Report
Comments
Authors have presented an article on “Appealing Renewable Materials in Green Chemistry” The manuscript in the current state can be accepted for publication after minor modifications mentioned below:
- The way of writing seems to be naive. There are numerous grammatical and scientific errors throughout the manuscript. Therefore, the language must be improved
- The information stated in scheme 1 is not justifiable with the reaction scheme and entry table. Kindly modify it to make it better.
- There are several references which are not in format such that the page no. and volume are missing, such as Reference 6, 17, 24, 26, 37, 53, 55, 56, 57, 59, 85, 94. Kindly update.
- There must be a separate list of abbreviations used in the paper.
- The figures and schemes used in the manuscript should be inculcated in tabular
- Following articles can be Cited:
- Materials Today Chemistry 8 (2018) 56-84
- Fibers and Polymers 20.4 (2019) 739-751.
After these changes, the article can be considered for publication.
Author Response
We thank the reviewers for their thorough review and precious comments and suggestions, which significantly improved the first version of the manuscript
- The authors revised the manuscript and straightened out several typos, trying to control any wording errors better. The manuscript was linguistically revised as suggested.
- The manuscript has been revised along the lines suggested by the reviewer. We have also added all the suggested references.
- Regarding point 5 raised by the reviewer, we have followed the Molecules Guidelines. Later, Figures and Schemes will be available to the editorial office.
- As requested by reviewers 4 and 5, a list of acronyms has been included at the end of the manuscript.
Reviewer 5 Report
This manuscript embodies a general overview of the raw materials (wool; silk; feathers; plants derivatives), once considered as a waste, now rethought as media for the extraction of pollutants, the adsorption of metals, and organic molecules for heterogeneous catalysis. The authors tried to help a deeper understanding of how green chemical processes can be terrifically performed and the use of these natural products for organic reactions, the extraction of pollutants, the adsorption of metals and organic molecules. Thus, this referee recommends this manuscript is suitable for publication in the Molecules.
Author Response
We thank the reviewer for reconsidering his/her decision on manuscript publication towards acceptance. We believe this study can significantly impact the chemical industry and academia, encouraging the adoption of a greener process that eliminates solvent use and reduces hazardous reagents.
Round 2
Reviewer 2 Report
The authors have now revised the manuscript carefully according to the comments. Also, some reasons to keep the organization of the original manuscript given by authors are acceptable.